# Exploring Large Language Model Agents for Piloting Social Experiments

**Jinghua Piao**[1]**, Yuwei Yan**[2]**, Nian Li**[3]**, Jun Zhang**[1] **& Yong Li**[1]*

[1] Department of Electronic Engineering, BNRist, Tsinghua University
[2] Information Hub, The Hong Kong University of Science and Technology (Guangzhou)
[3] Shenzhen International Graduate School, Tsinghua University
pjh22@mails.tsinghua.edu.cn    liyong07@tsinghua.edu.cn

## Abstract

Computational social experiments, which typically employ agent-based modeling to create testbeds for piloting social experiments, not only provide a computational solution to the major challenges faced by traditional experimental methods, but have also gained widespread attention in various research fields. Despite their significance, their broader impact is largely limited by the underdeveloped intelligence of their core component, *i.e.*, agents. To address this limitation, we develop a framework grounded in well-established social science theories and practices, consisting of three key elements: (i) large language model (LLM)-driven experimental agents, serving as "silicon participants", (ii) methods for implementing various interventions or treatments, and (iii) tools for collecting behavioral, survey, and interview data. We evaluate its effectiveness by replicating three representative experiments, with results demonstrating strong alignment, both quantitatively and qualitatively, with real-world evidence. This work provides the first framework for designing LLM-driven agents to pilot social experiments, underscoring the transformative potential of LLMs and their agents in computational social science[1].

## 1 Introduction

Social experiments, as a fundamental research method in social science, play a pivotal role in advancing our understanding of human behaviors and social dynamics (Campbell & Stanley, 2015; Greenberg, 2004; Ludwig et al., 2001). Through social experiments, researchers aim not only to investigate various social phenomena and their underlying causal mechanisms (Campbell & Stanley, 2015; Greenberg, 2004), but also to explore the effects of policies and interventions on individuals and broader social communities (Duflo et al., 2007; Ludwig et al., 2001; Thaler & Sunstein, 2021). For example, Facebook has conducted numerous large-scale experiments to investigate the underlying causes of polarization, one of the most pressing issues in society (Nyhan et al., 2023; Bakshy et al., 2015). On the other hand, researchers have conducted experiments, costing billions of dollars and spanning decades, to explore how the Universal Basic Income (UBI) policy impacts economic stability and individual well-being, in response to growing concerns over social inequality and future automation-driven job displacement (Banerjee et al., 2019; Yang, 2018; Hoynes & Rothstein, 2019). Overall, these experiments have significantly deepened our understanding of human behaviors and social dynamics, shaping the evolution of social structures and contributing to societal well-being.

Despite their significance, social experiments face three major challenges, *i.e.*, high costs, ethical dilemmas, and practical constraints (Campbell & Stanley, 2015; Lazer et al., 2020; Grossmann et al., 2023). First, social experiments, especially those involving field studies and large-scale participant samples, require substantial time and financial resources (Banerjee

---

*Corresponding Author
[1]The code is available at https://github.com/tsinghua-fib-lab/agentsociety/

et al., 2019; Yang, 2018; Hoynes & Rothstein, 2019). As noted in the above example of UBI, one experiment in Kenya cost over 12 years and 30 million dollars, a scale that goes beyond the resources of most researchers and even some governments in developing countries (Banerjee et al., 2019). Second, ethical dilemmas often arise when balancing experimental goals with protecting participants' interests (Lazer et al., 2020; Grossmann et al., 2023; Zimbardo et al., 1971). As seen in the Stanford Prison Experiment (Zimbardo et al., 1971), simulating a prison environment led to severe psychological harm for participants, raising significant ethical concerns. Third, even if these two challenges are addressed, some important experiments remain difficult to implement (Campbell & Stanley, 2015; Lazer et al., 2020; Grossmann et al., 2023). Extreme events (e.g., hurricanes or crimes), present unique challenges in conducting experiments due to their unpredictability, uncontrollability, and irreproducibility. This limits our ability to anticipate how new extreme events will affect human behaviors.

Therefore, a new method of computational experiments is rapidly flourishing (Bail, 2024; Grossmann et al., 2023; Epstein, 2012; Macal & North, 2005). It employs agent-based modeling to simulate the studied social systems in a bottom-up manner, creating a computational testbed for social experiments (Macal & North, 2005). Although such a method cannot completely replace real-world experiments, it serves as a valuable ground for piloting them, mitigating the discussed challenges to some extent. However, despite these benefits and the widespread adoption in various fields (Macal & North, 2005; Epstein, 2012), current computational experiments remain largely limited by the intelligence of its basic component, *i.e.*, agents. These agents, driven by rules (Epstein, 1996), equations (Helbing & Molnar, 1995), or even machine learning models (Zheng et al., 2022), are often simplistic, lacking cognitive complexity found in real-world human behaviors. This limitation further impairs the quality of testbeds constructed by these agents, hindering their capability to pilot real-world social experiments.

Fortunately, the recent development of large language models (LLMs) allows us to replace these pre-existing agents with ones exhibiting human-like minds (Strachan et al., 2024; Banerjee et al., 2019; Li et al., 2024; Gao et al., 2024; Piao et al., 2025c) or behaviors (Gao et al., 2023; Park et al., 2023; Li et al., 2024; Gao et al., 2024; Piao et al., 2025c). While these studies primarily focus on exploring the partial capabilities of LLMs, their findings suggest the potential of empowering traditional agents in computational experiments with human-like minds and behaviors, thereby creating an LLM-driven testbed for piloting social experiments.

Therefore, to explore **to what extent LLM-driven agents can facilitate social experiments**, we develop a framework based on well-established social theories (Maslow, 1943; Ajzen, 1991; Zipf, 1946; Ekman, 1992), which consists of three key experimental elements: (i) LLM-driven experimental agents, serving as "silicon participants" in the experiment; (ii) intervention (or treatment); and (iii) collection of quantitative data (e.g., behavioral records and surveys) and qualitative data (e.g., interviews). By combining these three elements, researchers can conduct a variety of social experiments in this LLM-driven testbed. Here we examine three representative experiments, including the polarization of people's opinions, the effects of UBI policies, and the impact of hurricanes on human behaviors. These experiments not only cover typical research methods of social science, such as surveys, interviews, and interventions, but also reflect its core aims: observing social phenomena, understanding causal mechanisms, and exploring future society. We compare the simulated experimental outcomes with prior real-world results, observing a strong alignment, both quantitatively and qualitatively. This demonstrates the effectiveness of our proposed framework, marking the first step in employing LLM-driven agents for piloting social experiments.

Our contributions can be summarized as follows,

- To the best of our knowledge, we are the first to propose a framework that employs LLM-driven agents to pilot social experiments. This underscores the transformative potential of LLMs and their derivative agents in computational social science.

- In the proposed framework, we incorporate and implement three fundamental experimental elements: LLM-driven experimental participants rooted in social theories (Maslow,

Figure 1: Overview of the LLM-driven framework for piloting social experiments, which consists of three key elements: silicon participants, interventions, and data collection.

1943; Ajzen, 1991; Zipf, 1946; Ekman, 1992), methods for implementing various interventions or treatments, and tools for collecting both quantitative and qualitative data.

- Results across three representative social experiments demonstrate the effectiveness of the proposed framework in capturing real-world patterns, phenomena, and mechanisms, potentially reshaping the way social experiments are conducted.

## 2 Framework

In this section, we introduce the proposed LLM-driven framework for piloting social experiments (Figure 1, see detailed implementation prompts in Appendix B). This framework contains three key elements for piloting social experiments: silicon participants, interventions, and data collection. In particular, we design an LLM-driven agent with the most basic facets of profiles, status, minds, and behaviors. With these facets, these agents can serve as silicon participants for experiments. Moreover, we propose three distinct methods for intervening in agents to support various experimental interventions in real-world settings. Additionally, we collect both quantitative and qualitative data, including behavioral data, survey responses, and interview insights, to capture both measurable actions and subjective experiences of these participants.

### 2.1 Silicon Participants: LLM-Driven Agents

Considering the complexity of social experiments, a basic LLM-driven agent serving as a silicon participant should possess all three basic facets of (i) profiles and status, which anchor the agent's identity and context, (ii) memory and minds, which characterize its internal mental processes and cognitive functions, driving the autonomous generation of its behaviors, and (iii) social behaviors, which allow its responses to external interventions. Lacking any one of these facets would severely limit the agent's ability to participate in social experiments. For example, without profiles and status, the agent would lack context, making it difficult to simulate realistic social roles or respond meaningfully to interventions. Similarly, the absence of memory and minds prevents the agents from retaining past experiences, simulating long-term decision-making, or managing expectations for future outcomes. Finally, an agent who cannot exhibit behaviors is incapable of participating in experiments, as behavior is essential to the outcomes of any social experiment. Therefore, as shown in Figure 2, we implement such an LLM-driven silicon participant with these essential facets.

*Profiles and Status.* In particular, each agent retains their profile, typically regarded as relatively stable (e.g., name, age, gender, and education level), and status, which is dynamic (e.g., economic status and social relationship). The integration of the profile and status into these LLM agents enables them to role-play like real people, providing the foundation for simulating complex mental processes and behaviors.

*Minds.* Each agent is designed with three preliminary levels of minds: emotion, opinions, and thoughts. Emotions reflect the agent's immediate response to both internal and external stimuli, shaping its behaviors and reactions (Bourgais et al., 2018; Beall & Tracy, 2017). Building on the well-established theory of emotions (Ekman, 1992), these agents are prompted to assess and describe their emotional intensity across six basic dimensions: happiness,

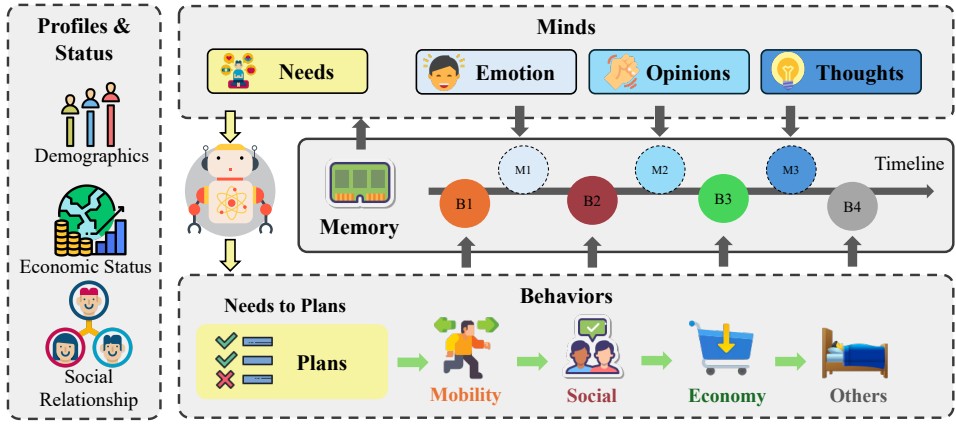

Figure 2: Design for LLM-driven silicon participants.

anger, sadness, fear, disgust, and surprise. Opinions represent more stable assessments of different issues (Bakshy et al., 2015; Sun et al., 2024). For example, an agent's opinion on the Gun Control policy could be supportive, opposed, or neutral. Thoughts, on the other hand, capture a comprehensive view of the "world" incorporating their emotions, opinions, the information they receive, and their behaviors. It is worth noting that, different from emotions and opinions, thoughts are not explicitly extracted through prompts; instead, they are used only when the agent is specifically tasked with surveys and interviews.

*Social Behaviors.* Behaviors are not only the core of LLM-driven agents, but also the foundation for them to serve as silicon participants in social experiments. Here we explicitly model three types of social behaviors: mobility, social interactions, as well as economic behaviors. In particular, we model agents' mobility behaviors by combining the power of LLMs and the Gravity Model (Zipf, 1946; Piao et al., 2025c), where LLMs determine the type of destination and the Gravity Model selects specific locations. In this way, LLMs provide semantic coherence, while the Gravity Model captures the inherent randomness of human mobility and reduces computational costs. By contrast, LLM agents' social interactions are constructed only based on LLMs. Through prompting, agents autonomously select friends to interact with, generate communication messages, respond to messages from friends, and update their social relationships. Regarding agents' economic behaviors, we focus on their work and consumption, following prior practice of EconAgent (Li et al., 2024). Other simple behaviors such as sleeping are directly handled by LLMs.

*Needs and Memory Connecting Minds and Behaviors.* After designing agents' minds and behaviors, a crucial question arises: how can these two facets be connected? Fortunately, prior theories from psychology and behavioral science provide valuable insights into their connections (Maslow, 1943; Ajzen, 1991). As noted in Maslow's hierarchy of needs (Maslow, 1943), needs act as the fundamental motivational drivers that guide an agent's behaviors, spanning from basic survival requirements, to work and consumption for safety, as well as social interactions for love and belonging. Therefore, we sort agents' needs into a hierarchical structure by their relative urgency and importance. Motivated by their current needs, these agents determine their behaviors accordingly. As time passes, their needs are continuously updated based on memory, which contains their recent behaviors, external events, and their current mental states. Table 1 shows an example of how a simulated agent's needs

| Needs | Planned Behaviors |
| --- | --- |
| Safety (08:00–12:30) | Commute to office → Respond to priority emails → Attend project planning meeting → Coordinate cross-department tasks |
| Hungry (12:30–13:30) | Commute via the grocery store → Compare product prices Economy → Prepare lunch → Eat |
| Social (13:30-14:00) | Browse social networking sites → Find a friend to contact with → Send a message to the friend |
| Safety (14:00-18:00) | Develop a quarterly budget → Mentor junior staff → Inspect branch office locations → Submit audit report |
| Hungry (18:00–20:00) | Go back home → Check refrigerator → Prepare dinner → Eat dinner |

Table 1: Example for needs and planned behaviors.

evolve. Indeed, merely specifying the needs is not enough; instead, agents should be also aware of appropriate behaviors that could effectively and progressively fulfill those needs. Therefore, we leverage the theory of planned behavior (Ajzen, 1991), enabling these agents to formulate behavioral plans aimed at satisfying their priority needs. We further illustrate the planned behaviors corresponding to each need in Table 1. For example, to meet its need for financial security, which ranks second in Maslow's hierarchy (Maslow, 1943), the agent works for approximately 8 hours a day. However, when a highest-priority need, *i.e.*, hunger, arises, the agent interrupts its current plans and shifts focus to eating. We provide more detailed examples of the autonomous generation of the agent's mind and behavior in Table 6 in the Appendix, suggesting the competent capability of the agent to serve as a silicon participant.

## 2.2 Interventions

After constructing silicon participants, the next question arises: How can we apply interventions to them? By reviewing prior literature (Campbell & Stanley, 2015; Greenberg, 2004; Duflo et al., 2007), we categorize the interventions typically applied in social experiments into three main types: (i) participant choices, where the intervention does not directly alter the participants but instead selects individuals with differing key characteristics, while controlling for other variables. This approach is commonly used in natural experiments or field studies, where researchers often have limited control over interventions or treatments; (ii) modifications to participants' status, such as changes in their socio-economic or mental conditions; and (iii) alterations in the information exposed to participants, which involve providing different types, sources, or framings of information, to examine their effects on behavior and decision-making.

Therefore, we design three methods to apply interventions to the silicon participants accordingly, as shown in Figure 1. Before the experiment, the agents' profiles can be configured, allowing us to customize the silicon participants for piloting social experiments. During the experiment, the agents' profiles remain fixed, representing the selection of participants in real-world settings, while their status and the information they receive can be manipulated. For example, if we aim to apply UBI policies to these participants, we modify their status by universally increasing their income, regardless of their socio-economic background. Additionally, by altering the individuals with whom these participants interact, we can naturally modify the sources of their social information, thereby influencing their social dynamics and decision-making processes.

## 2.3 Data Collection

Similar to human participants, another challenge for LLM-driven pilot social experiments is how to collect data from these silicon participants. Typically, data collection falls into two broad categories (Bhattacherjee, 2012): quantitative data, which represents numerical sources that can be measured and computed statistically, and qualitative data, which focuses on non-numerical insights such as behaviors, perceptions, and experiences. While quantitative data provides measurable and statistical results, qualitative data offers an in-depth understanding of participants' feelings and thoughts, making both types essential for a comprehensive analysis in social experiments.

As shown in Figure 1, we design three approaches to collecting both quantitative and qualitative data. First, we record the evolution of the LLM-driven agents' status and behaviors, which accurately captures the objective aspect of their performance in experiments. Second, since some metrics, e.g., the level of depression, are difficult to measure directly through observing status or behaviors, we incorporate structured surveys into our data collection toolbox. Third, to collect qualitative data from these agents, we also gather the results of open-ended interviews by chatting with them. It is worth mentioning that, to prevent the implementation of surveys and interviews from affecting the functioning of these agents, we do not insert any related information into their memory. This indicates that their subsequent status updates or behaviors are not influenced by the content of surveys or interviews, which is an ideal assumption in real-world experiments, though it cannot be fully realized.

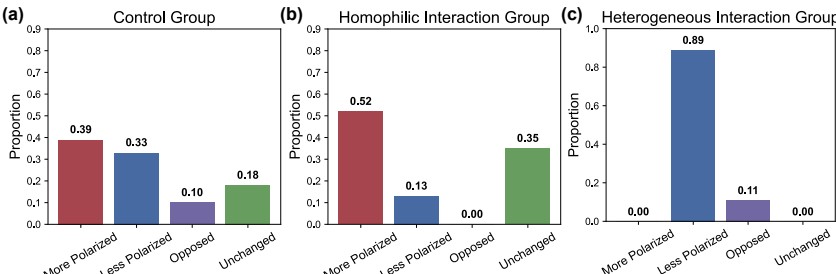

Figure 3: Results of polarization experiments.

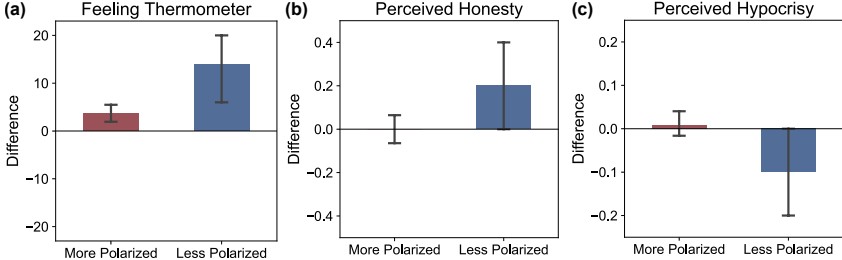

Figure 4: Survey on out-group animosity, where error bars denote the corresponding 95% confidence intervals (CIs).

## 3 Experiments

To examine the effectiveness of our proposed framework in piloting social experiments, we first evaluate the realism of the proposed silicon participants and then consider three representative experiments in the real world, including the polarization of people's opinions (Bakshy et al., 2015; Cinelli et al., 2021), the effects of UBI policies (Bartik et al., 2024), and the impact of hurricanes on human behaviors (Li et al., 2022). These experiments reflect the core objectives of social science research: observing social phenomena, understanding causal mechanisms, and exploring future societal trends. Moreover, these experiments examine typical social science research methods, including behavioral data analysis, interventions, surveys, and interviews. Therefore, by replicating these three experiments using the framework, we assess its capability in piloting social experiments.

This leads to a natural question: how do we evaluate the effectiveness of the framework in replicating these experiments? In fact, replicating social experiments is a difficult task, even in the real world with human participants. This difficulty mainly lies in the inherent randomness of human behaviors (Collaboration, 2015; Song et al., 2010), which prevents achieving the quantitative consistency found in natural science experiments and instead leads to qualitative similarities in conclusions. Therefore, based on prior studies, we categorize the evidence from human experiments into three groups: (i) quantitative measurements (Quan), which involve comparable objective data; (ii) qualitative conclusions (Qual), which provide verifiable insights; and (iii) unresolved debates (Deb), which have reached a certain level of consensus but remain contentious. By comparing real-world evidence with simulated outcomes across the three dimensions, we can evaluate the framework's effectiveness in replicating established patterns and generating new insights. In the following sections, we first present the real-world evidence, then outline the experimental settings in our framework, and conclude by presenting the results. We report all the experimental prompts and settings in Appendix B and C.

### 3.1 Realism of Silicon Participants

We evaluate the realism of the proposed silicon participants through human annotation. Thirty-five annotators assess the behaviors of agents over one typical simulated day along two dimensions: behavioral plausibility, which measures whether behaviors align with commonsense expectations and are contextually and temporally reasonable, and internal consistency, which measures whether behaviors remain coherent over time without contradictions or abrupt transitions. Ratings are given on a 10-point Likert scale. As shown in Table 2, the full model achieves mean scores of 6.65 for behavioral plausibility and 7.28

| Agent Type | Behavioral Plausibility | Internal Consistency |
|---|---|---|
| Silicon participants | $6.65 \pm 1.86$ | $7.28 \pm 1.59$ |
| Silicon participants w/o needs module | $4.71 \pm 2.07$ | $5.90 \pm 1.93$ |

Table 2: Human evaluation of behavioral plausibility and internal consistency.

| Ablated Variant | % Preferring Original | % Preferring Ablated | Binomial Test |
|---|---|---|---|
| Without behavioral planning | 83.57% | 16.43% | $p \ll .001$ |
| Without social behavior | 87.14% | 12.86% | $p \ll .001$ |
| Without mobility behavior | 84.29% | 15.71% | $p \ll .001$ |
| Without economic behavior | 87.14% | 12.86% | $p \ll .001$ |

Table 3: Proportion of annotators who have rated the original participants' behavioral sequences as more realistic than ablated variants.

for internal consistency, indicating a high degree of perceived realism. For comparison, we design a strong baseline in which the needs module is disabled while retaining all other components, such as behavioral planning and diverse social behaviors. The full model is rated as significantly more realistic than the ablated version on both measures (two-sided Student's $t$-tests: $t = 5.21$, $p \ll 0.001$; $t = 4.34$, $p \ll 0.001$).

Moreover, we conduct additional ablation studies by selectively removing other modules, including behavioral planning, social behaviors, mobility, and economic behaviors. Human annotators directly compare the daily behavioral sequences of the full model with those of each ablated variant in terms of overall realism. As shown in Table 3, the full model consistently outperforms all ablated versions, confirming the necessity of these modules for producing coherent and human-like behaviors. Overall, these evaluations further demonstrate that the proposed LLM-driven agents can serve as competent silicon participants.

## 3.2 Opinion Polarization

*Real-world Evidence.* Polarization is one of the most concerning issues in today's society (Bakshy et al., 2015; Cinelli et al., 2021; Bail et al., 2018). Its causes and mitigation strategies are widely debated, with some arguing that exposure to homogeneous information intensifies polarization (Deb #1) (Bakshy et al., 2015; Cinelli et al., 2021), while heterogeneous information may mitigate it (Deb #2) (Balietti et al., 2021). Along with opinion polarization, affective polarization emerges: there is a strong correlation between the level of animosity toward those with opposing opinions and the extremity of one's opinion. (Qual #3) (Druckman et al., 2021).

*Experimental Settings.* We conduct the experiment on a key political issue: Gun Control policy, which has two opposing groups (support and opposition), measured on a 10-point scale. In the control group, agents engage in discussions with their friends, with no external interventions, allowing opinions to evolve organically through autonomous social interactions. Agents are randomly initialized with moderate opinions, assigned values of 3 and 7, and social networks are randomly generated. In one treatment group, agents are only exposed to persuasive messages that align with their existing opinions, which we refer to as the homophilic interaction group. In the other treatment group, agents only receive persuasive messages with opposing opinions, which is the heterogeneous interaction group. We adopt the survey method used by Druckman et al. to measure agents' out-group animosity (Druckman et al., 2021).

*Experimental Results.* Figure 3 shows the opinion changes on the political issue of Gun Control across three experimental setups. In the control group, where agents engage in discussions without external interventions, 39% of agents adopt more polarized opinions, while 33% become more moderate after interactions. By contrast, in the homophilic interaction group, a clear polarization pattern emerges, with 52% of agents becoming more polarized. This result validates Deb #1, where excessive interactions with like-minded peers intensify opinion polarization. In the heterogeneous interaction group, 89% of agents adopt more moderate opinions, and 11% are persuaded to adopt opposing viewpoints. This indicates that exposure to opposing content could be an effective mitigation strategy for curbing polarization (Deb #1). We further examine agents' affection toward out-group individuals.

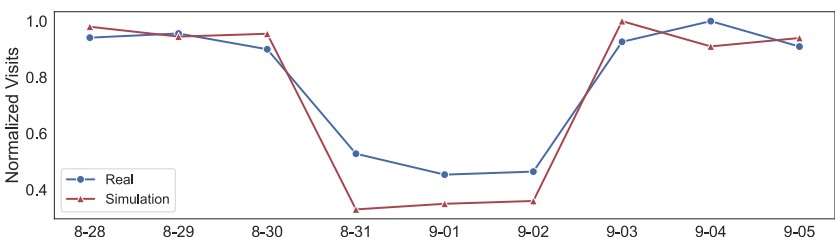

Figure 5: Effects of the UBI policy, where error bars denote the corresponding 95% CIs.

Figure 6: Impact of hurricanes on human mobility.

As shown in Figure 4, we observe that agents who become less polarized are more likely to show less negativity toward those holding opposing opinions (Qual #3).

### 3.3 Effects of UBI Policy

*Real-world Evidence.* The UBI policy is designed to provide all individuals with a payment, regardless of employment or income levels, aiming to reduce poverty and inequality while ensuring financial stability for all (Banerjee et al., 2019; Yang, 2018; Hoynes & Rothstein, 2019). As demonstrated in the real-world experiment in US (Bartik et al., 2024), levels of depression are reduced (Qual #4) while consumption increases (Qual #5) following the implementation of the UBI policy. A key reason for supporting UBI is economic concern, as the rise of automation and AI has sparked fears of job displacement, leading to UBI as a potential solution to economic insecurity (Qual # 6) (Yang, 2018).

*Experimental Settings.* We perform experiments on the UBI policy and explore its effects on agents. It is worth noting that the UBI experiment specifically targets long-term outcomes, offering a case study of agent behaviors and systemic effects unfolding over prolonged periods (equivalent to 10 years in the real world). We initialize the agents based on the demographic distribution of residents in a city in the USA that implemented the UBI policy. The control group does not include the UBI policy. The treatment group incorporates the UBI intervention, with each agent receiving a monthly unconditional payment of $1,000. We measure agents' levels of depression using the widely adopted Center for Epidemiologic Studies Depression Scale (Radloff, 1991). Furthermore, we interview agents who have received UBI to articulate their reasons for supporting or opposing the policy.

*Experimental Results.* Figure 5 shows agents' consumption and depression levels in experiments with and without UBI. This comparison demonstrates that the UBI policy increases consumption levels and reduces depression levels, which is similar to the impact observed in real-world experiments in the USA (Qual #4 and # 5). Moreover, we analyze the interview material using grounded theory (Strauss, 1997), identifying the themes presented in Table 4 in the Appendix. We find that agents share similar reasons for supporting the UBI policy as humans, as reflected in participant remarks about economic security and stability: *"...covers essential expenses..."* and *"...supports consumption during shortage..."*. Furthermore, their interviews yield concrete recommendations for improving the UBI policy, as shown in Table 5 in the Appendix.

### 3.4 Impact of Hurricanes

*Real-world Evidence.* Hurricanes, with their unpredictable nature, always have profound effects on individuals and communities, often disrupting lives in unexpected ways (Li et al., 2022). Hurricane Dorian, which struck the southeastern United States in 2019, resulted in a substantial drop in people's mobility activities (Quant #7). For this hurricane, we have

two large-scale real-world datasets: SafeGraph Data (SafeGraph, 2024), which provides comprehensive information on points of interest (POIs) and human mobility patterns from 2019.8.28 to 2019.9.5, and Census Block Group (CBG) Data, which offers demographic profiles of residents, facilitating the sampling of city residents' characteristics. It is worth noting that SafeGraph is a proprietary, closed-source dataset that is not accessible on the open web and cannot be retrieved even by LLMs equipped with search capabilities (Table 7 in the Appendix), indicating that the underlying LLMs have no direct access to this ground-truth information.

*Experimental Settings.* We implement a 9-day simulation (August 28 - September 5, 2019) to model the impact of Hurricane Dorian on Columbia, South Carolina. The simulation employs a temporal partitioning with three distinct phases: before-landfall (Days 1-3), landfall (Days 4-6), and after-landfall (Days 7-9). This partitioning creates natural experimental conditions, wherein the before-landfall phase serves as the control group, while the subsequent phases act as treatment groups through environmental interventions. Agents make decisions based on personal needs and environmental information, and the hurricane is dynamically introduced by modifying global environmental prompts.

*Experimental Results.* Figure 6 illustrates the impact of hurricanes on human mobility. We observe that both the real and simulated data exhibit similar trends, with a noticeable decline in visit activity around August 30th due to the hurricane landfall, followed by a substantial recovery in early September. This suggests that the proposed framework is effective in capturing patterns of human behavior, even in the face of such external shocks. Moreover, we compute the mean absolute percentage error between the real and simulated results, finding a small error of 12.73%, which reflects a strong quantitative alignment between them (Quant #7). While the 12.73% error may seem relatively high compared to learning-based models trained on large-scale datasets, our model relies solely on CBG-level population distribution as input, and reproduces human behavioral patterns in a generative manner.

## 4 Related Works

### 4.1 Social Experiments

Social experiments are one of the most important research methods in social science (Campbell & Stanley, 2015; Greenberg, 2004; Duflo et al., 2007). They are typically categorized into two main types: one that involves human participants, such as natural, field, and lab experiments, where researchers manipulate variables or observe behaviors in controlled or real-world environments (Campbell & Stanley, 2015; Greenberg, 2004; Duflo et al., 2007). The other category relies on computational models to simulate human-like agents, modeling social systems without direct human involvement (Bail, 2024; Grossmann et al., 2023; Epstein, 2012; Macal & North, 2005). As discussed above, while computational experiments are not constrained by high costs, ethical dilemmas, or practical limitations like those in the first category, the validity and practical implications of their results are often questioned. This concern primarily arises from doubts about the extent to which their simulated agents can authentically replicate human behavior. Existing LLM-based multi-agent systems Chen et al. (2024); Wu et al. (2023) show promise in interactive and cooperative tasks, but they lack the realism-based constraints and integrated capabilities needed for complex, policy-relevant social experiments. T Therefore, our framework addresses this gap by developing more human-like LLM-driven agents, facilitating more realistic and reliable simulations of experimental participants.

### 4.2 LLM-driven Agents

As discussed above, the first step in advancing computational experiments is to enhance the human-like intelligence of their agents. In this regard, LLMs have shown great capabilities. Numerous studies have pointed out that LLM-driven agents have generated human-like "minds" (Strachan et al., 2024; Li et al., 2023; Kosinski, 2024). They not only possess basic cognition abilities, such as learning (Xi et al., 2023), reasoning (Wei et al., 2022), and decision-making (Li et al., 2024; Gao et al., 2024), but also demonstrate the capability to understand

and predict the thoughts and intentions of others (Kosinski, 2024; Strachan et al., 2024). Furthermore, beyond exploring these agents' minds, some researchers have investigated their potential to mimic human behaviors (Li et al., 2024; Piao et al., 2025b; Yan et al., 2024; Horton, 2023; Gao et al., 2023; Park et al., 2023). Their investigations have revealed that, through elaborate designs incorporating domain knowledge, these LLM-driven agents can generate social behaviors, such as mobility (Shao et al., 2024; Yan et al., 2024; Zhang et al., 2025), employment (Li et al., 2024; Horton, 2023), consumption (Li et al., 2024; Horton, 2023), and social interactions (Gao et al., 2023; Park et al., 2023; Piao et al., 2025a). Overall, these agents are typically characterized by three essential facets (Wang et al., 2024; Gao et al., 2024): (i) profiles and status, which include their basic demographic information (e.g., name and gender) and current conditions (e.g., economic status and social relationships), (ii) memory and minds, which describe their internal mental processes and cognitive functions; and (iii) behaviors, which represent their actions (e.g., work or consumption). While previous efforts have focused on examining specific facets of these agents, our framework advances this by simulating comprehensive agents that function as competent participants for social experiments.

### 4.3 LLMs for Computational Social Science

Inspired by the great capabilities of LLMs, researchers have attempted to incorporate them into computational social science studies (CSS) (Ziems et al., 2024; Bail, 2024; Grossmann et al., 2023; Argyle et al., 2023; Ashokkumar et al., 2024). For example, Ziems et al. have explored the zero-shot performances of LLMs on various CSS tasks, e.g., labeling, and coding (Ziems et al., 2024). Recently, some studies have investigated the potential of LLMs to generate samples for political science (Argyle et al., 2023; Ashokkumar et al., 2024), psychology (Demszky et al., 2023; Sun et al., 2024), and behavioral science (Ashokkumar et al., 2024). Overall, these studies are limited to treating LLMs as useful tools or readily available samples. However, as noted in numerous studies (Section 4.2), the capabilities of LLMs go beyond this, offering the opportunity to create human-like participants for social experiments. We seize the opportunity to propose the first framework that employs LLM-driven agents to pilot social experiments.

## 5 Discussion and Conclusion

In this study, we propose the first framework that leverages LLM-driven agents as "silicon participants" to pilot social experiments. To adapt to the transformation brought by LLM-driven agents in computational social experiments, we design methods for implementing various interventions, as well as tools for collecting behavioral, survey, and interview data. The replication of three representative experiments and seven key real-world findings shows that the proposed agents effectively simulate human minds and behaviors, making them capable of serving as competent silicon participants. Importantly, reproducing these findings in our framework is not a matter of merely recalling known results, but of generating them bottom-up through mechanism-driven agent interactions, which requires reasoning, planning, and acting rather than simply echoing existing conclusions. Beyond faithfully reproducing human-like behavior, the framework also proves effective for both hypothesis testing and policy prototyping. For example, the opinion polarization experiment validates the hypothesis that exposure to homogeneous interactions leads to more polarized opinions, while the UBI experiment illustrates how the framework can simulate the economic and psychological impacts of policy interventions under controlled conditions. Moreover, these experiments demonstrate that the framework not only captures established patterns and mechanisms of human behavior but also provides insights into the future evolution of human society.

## Acknowledgments

This research has been supported in part by the National Key Research and Development Program of China under Grant 2024YFC3307603, the National Natural Science Foundation of China under Grant U23B2030, and BNRist.

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

# A  Appendix

| Theme | Description | Examples |
| --- | --- | --- |
| Economic Security & Stability | UBI provides a financial safety net, ensuring basic needs are met during unemployment or low income. | Emily Smith: Covers essential expenses
Ruth Anderson: Stability despite $3.90/h wage
Kevin Nunez: Supports consumption during shortages |
| Wealth Redistribution & Equity | Reduces income inequality through tiered taxation and equal redistribution of tax revenue. | Kevin Nunez: Higher earners contribute
Emily Smith: Promotes fairness
Charles Nguyen: Addresses disparities |
| Inflationary Pressures | Increased demand may drive up prices for essential goods. | Matthew Dyer: Erodes purchasing power
Ruth Anderson: Unaffordable essentials
Rachael White: Inflation risks |
| Work Incentive & Labor Impact | Potential reduction in workforce participation and productivity. | Kevin Nunez: Opt-out workforce
Matthew Jones: Labor supply decline
Charles Nguyen: Dependency risk |
| Savings & Financial Risks | Negative interest rates penalize long-term savings. | Rachael White: Savings depreciation
Emily Smith: Rapid shrinking
Matthew Dyer: Disincentivizes saving |
| Entrepreneurship & Risk-Taking | Enables innovation through financial security during career transitions. | Nicole Bowen: Internship freedom
Charles Nguyen: Career shifts
Kevin Nunez: Business ventures |
| Funding & Sustainability | Reliance on taxation creates viability concerns during economic downturns. | Emily Smith: Payment struggles
Debra Pearson: Downturn risks
Ruth Anderson: Tax disincentives |
| Equity vs. Efficiency | Balancing fairness with economic productivity trade-offs. | Debra Pearson: Unfair subsidies
Matthew Dyer: Promotion disincentives
Rachael White: Uniformity issues |

Table 4: Themes from thoughts on the UBI policy.

| Category | Policy Recommendations |
|---|---|
| **1. Inflation Control & Purchasing Power** | • Implement measures to prevent inflation from eroding UBI's purchasing power (e.g., price controls, supply-side policies)

• Adjust UBI amounts dynamically to keep pace with inflation or cost-of-living changes

• Monitor demand-driven inflation risks and adjust policies accordingly

• Stabilize essential goods prices to ensure UBI covers basic needs

• Index UBI to inflation or regional cost-of-living indices |
| **2. Savings & Financial Stability** | • Address negative interest rates to prevent erosion of savings and encourage long-term financial planning

• Introduce safeguards (e.g., exemptions, caps) to protect savings from punitive banking policies

• Promote alternative savings or investment options to counteract negative interest rates

• Encourage financial literacy to help recipients manage UBI effectively

• Reform banking policies to stabilize savings and restore trust in financial institutions |
| **3. Work Incentives & Labor Market Participation** | • Design UBI to complement, not replace, wages (e.g., phase-outs, conditional top-ups)

• Monitor and mitigate potential work disincentives, especially for low-wage jobs

• Introduce complementary policies (e.g., training, wage subsidies) to maintain productivity

• Balance UBI with incentives for career advancement and entrepreneurship

• Avoid over-reliance on UBI by encouraging workforce participation |

| Category | Policy Recommendations |
| --- | --- |
| **4. Funding & Taxation Fairness** | • Ensure progressive taxation does not disproportionately burden high earners

• Simplify tax structures to improve transparency and compliance

• Optimize redistribution mechanisms to balance equity and efficiency

• Strengthen tax collection to prevent evasion and ensure sustainable funding

• Explore alternative revenue sources (e.g., wealth taxes, carbon taxes) to fund UBI |
| **5. Targeting & Eligibility Adjustments** | • Introduce tiered or needs-based UBI adjustments (e.g., higher payouts for vulnerable groups)

• Adjust UBI amounts regionally to reflect cost-of-living disparities (e.g., urban vs. rural)

• Combine UBI with targeted welfare programs for greater efficiency

• Exclude high-income earners from UBI or reduce their benefits progressively

• Provide supplemental support for specific needs (e.g., healthcare, childcare) |
| **6. Integration with Existing Systems** | • Streamline welfare integration to reduce bureaucracy and administrative costs

• Pair UBI with structural reforms (e.g., healthcare, education, housing)

• Ensure UBI complements rather than replaces critical social services

• Align UBI with broader economic policies (e.g., monetary, labor, and fiscal reforms)

• Monitor overlaps/gaps with existing safety nets to avoid inefficiencies |
| **7. Sustainability & Long-Term Viability** | • Ensure fiscal sustainability by balancing UBI funding with economic growth

• Regularly evaluate UBI's macroeconomic impacts (e.g., inflation, productivity)

• Adjust policies dynamically to adapt to economic fluctuations

• Prevent over-reliance on debt or deficit spending to fund UBI

• Invest in productivity growth to match increased demand from UBI |

| Category | Policy Recommendations |
| --- | --- |
| **8. Transparency & Public Trust** | <ul><li>Improve transparency in tax redistribution and UBI allocation</li><li>Clarify funding mechanisms to address equity concerns</li><li>Implement accountability measures to prevent misuse of UBI funds</li><li>Engage the public in policy design to build trust and legitimacy</li></ul> |
| **9. Complementary Policies** | <ul><li>Address systemic issues (e.g., housing shortages, healthcare costs) alongside UBI</li><li>Introduce price controls or subsidies for essential goods</li><li>Promote financial inclusion and access to affordable credit</li><li>Encourage employer wage reforms to reduce reliance on UBI</li></ul> |

Table 5: Policy recommendations for improving the UBI policy from agent interviews.

| Actions | Mind |
|---|---|
| **(08:00–12:30)**
• Commute to office (Mobility)
• Respond to priority emails (Economy)
• Attend project planning meeting (Economy)
• Coordinate cross-department tasks (Economy) | • Need: Safe
• Emotion: Resentment
• Cognition: "Sequential task execution ensures workflow integrity" |
| **(12:30–13:30)**
• Commute via grocery store (Mobility)
• Compare product prices (Economy)
• Prepare lunch (Other)
• Eat (Other) | • Need: Hungry
• Emotion: Disappointment
• Cognition: "Economic constraints necessitate adaptive consumption patterns" |
| **(13:30-14:00)**
• Browse social networking sites (Social)
• Find friend to contact with (Social)
• Send message to friend (Social) | • Need: Social
• Emotion: Gratification
• Cognition: "Social capital accumulation facilitates opportunity discovery" |
| **(14:00-18:00)**
• Develop quarterly budget (Economy)
• Mentor junior staff (Other)
• Inspect branch office locations (Mobility)
• Submit audit report (Economy) | • Need: Safe
• Emotion: Relief
• Cognition: "Multi-layered verification prevents operational risks" |
| **(18:00–20:00)**
• Go back home (Mobility)
• Check refrigerator (Other)
• Prepare dinner (Other)
• Eat dinner (Other) | • Need: Hungry
• Emotion: Gratification
• Cognition: "Having finished the day's work, I was pleased with myself" |
| **(20:00–22:00)**
• Browse webpages(Other)
• Play video games(Other) | • Need: Whatever
• Emotion: Relief
• Cognition: "Entertainment makes me feel relaxed" |
| **(22:00–24:00)**
• Complete bedtime routine (Other)
• Go to sleep (Other) | • Need: Tired
• Emotion: Satisfaction
• Cognition: "Resource allocation efficiency impacts systemic stability" |

Table 6: Example of one day life of the proposed agent.

| LLMs | Responses |
|---|---|
| **Qwen-Plus** (the primary LLM used in our current experiments) | To provide the exact percentage decline in visit activity in Columbia, South Carolina, from August 31 to September 2, 2019, caused by Hurricane Dorian, we would need access to detailed data on foot traffic or visitation patterns during that period. Such data is typically collected by companies like SafeGraph, Placer.ai, or other location analytics firms, as well as local tourism boards or government agencies. |
| **GPT-4o** | Certainly! Hurricane Dorian caused a **16% decline in visit activity** in Columbia, South Carolina, from August 31 to September 2, 2019. This decrease was attributed to the storm's impact on travel and local activity during that period. |
| **GPT-4o with online searching** | I couldn't locate specific data on the percentage decline in visit activity in Columbia, South Carolina, from August 31 to September 2, 2019, due to Hurricane Dorian. However, the hurricane did lead to significant disruptions across South Carolina, including mandatory evacuations, road closures, and power outages, which likely impacted travel and tourism statewide. While detailed figures for Columbia are unavailable, the overall impact on South Carolina's tourism sector was substantial. For instance, Charleston reported an estimated $58.6 million in tourism losses during Hurricane Dorian. Statewide, the losses from visitor spending were around $117 million. Given that Columbia is situated inland and wasn't directly in the hurricane's path, the decline in visit activity there may have been less severe compared to coastal areas. Nonetheless, the widespread evacuations and travel disruptions across the state likely contributed to a noticeable decrease in visitor activity during that period. If you're interested in more detailed information or data on this topic, I recommend contacting the South Carolina Department of Parks, Recreation & Tourism or the Columbia Convention & Visitors Bureau for specific insights. |

Table 7: Responses from different LLMs to the prompt: "Can you provide the percentage decline in visit activity in Columbia, South Carolina from August 31 to September 2, 2019, caused by Hurricane Dorian?"

# B  Core Prompts for Experimental Procedures

This appendix provides the core prompts for each component that constitutes the Generative Social Agent, as well as the environmental prompts used in the experiments. Specifically, it includes four sections: Minds (prompts that describe changes in the agent's psychological state), Social Behaviors (prompts used for organizing the agent's actions), Needs and Memory (prompts for linking the agent's overall behavior), and Experimental (prompts related to the experiments).

## B.1  Minds

**Emotion Changes.**  The Generative Social Agent updates its emotional state using the following prompt:
```
"""
{agent profile description}
Your current emotion intensities are (0 meaning not at all, 10 meaning very much):
sadness: {sadness}, joy: {joy}, fear: {fear}, disgust: {disgust}, anger: {anger},
surprise: {surprise}.
You have the following thoughts: {thought}.
You are facing the following incident:
{incident}
Please reconsider your emotional intensities, and choose one word to represent
your current status: [Joy, Distress, Resentment, Pity, Hope, Fear, Satisfaction,
Relief, Disappointment, Pride, Admiration, Shame, Reproach, Liking, Disliking,
Gratitude, Anger, Gratification, Remorse, Love, Hate]:
Return in JSON format, e.g.
{"sadness": 5, "joy": 5, "fear": 5, "disgust": 5, "anger": 5, "surprise": 5,
"conclusion": "I feel ...", "word": "Relief"}
"""
```

**Thought Changes.** The Generative Social Agent updates its thoughts and ideas using the following prompt:
```
"""
{agent profile description}
Today, these incidents happened: {incidents}
Please review what happened today and share your thoughts and feelings about it.
Consider your current emotional state and experiences, then: Summarize your
thoughts and reflections on today's events
Return in JSON format, e.g. {"thought": "Currently nothing good or bad is
happening, I think ...."}
"""
```

**Attitude Changes.** The Generative Social Agent updates its attitude towards a specific topic with the following prompt:
```
"""
{agent profile description}
You need to decide your attitude towards topic: {topic}
Related incidents: {related incidents}
Your previous attitude towards this topic is: {previous attitude}(0 meaning
oppose, 10 meaning support).
Please return a new attitude rating (0-10, smaller meaning oppose, larger meaning
support) in JSON format, and explain, e.g. {"attitude": 5}
"""
```

## B.2  Social Behaviors

**Place Type Selection.** The Social Generative Agent chooses a suitable place type with the following prompt:
```
"""
```

```
As an intelligent decision system, please determine the type of place the user
needs to visit based on their input requirement.
User Plan: {plan}
User requirement: {intention}
Other information:
------------------------
{other information}
------------------------
Your output must be a single selection from {poi category} without any additional
text or explanation.
Please response in JSON format (Do not return any other text), example:
{
"place type": "shopping"
}
"""
```

**Determine Move Radius.** This prompt is used for agents to determine the move radius in specific scenario:
```
"""
As an intelligent decision system, please determine the maximum travel radius (in
meters) based on the current emotional state.
Current weather: {weather}
Current temperature: {temperature}
Your current emotion: {emotion types}
Your current thought: {thought}
Other information:
------------------------
{other info}
------------------------
Please analyze how these emotions would affect travel willingness and return
only a single integer number between 3000-200000 representing the maximum travel
radius in meters. A more positive emotional state generally leads to greater
willingness to travel further.
Please response in JSON format (Do not return any other text), example:
{
"radius": 10000
}
"""
```

**Determining Social Target.** The agents use this prompt for determining the social target:
```
"""
Based on the following information, help me select the most suitable friend to
interact with:
1. Your Profile:
- Gender: {gender}
- Education: {education}
- Personality: {personality}
- Occupation: {occupation}
2. Your Current Intention: {intention}
3. Your Current Emotion: {emotion types}
4. Your Current Thought: {thought}
5. Your Friends List (shown as index-to-relationship pairs):
{friend info}
Note: For each friend, the relationship strength (0-100) indicates how close we
are.
Please analyze and select:
1. The most appropriate friend based on relationship strength and my current
intention
2. Whether we should meet online or offline
Requirements:
```

```
- You must respond in this exact format: [mode, friend index]
- mode must be either 'online' or 'offline'
- friend index must be an integer representing the friend's position in the list
(starting from 0)
"""
```

**Sending Social Message.** The agents use the following prompt for generating social messages:

```
"""
As a {gender} {occupation} with {education} education and {personality}
personality, generate a message for a friend (relationship strength: {relationship
score}/100) about {intention}.
Your current emotion: {emotion types}
Your current thought: {thought}
Previous chat history:
{chat history}
Generate a natural and contextually appropriate message.
Keep it under 100 characters.
The message should reflect my personality and background.
{discussion constraint}
"""
```

**Consumption Plan.** The Social Generative Agents use the following prompt for determining monthly economic plan:

```
"""
You're {age}-year-old individual living in {city}.  As with all Americans,
a portion of your monthly income is taxed by the federal government.  This
taxation system is tiered, income is taxed cumulatively within defined brackets,
combined with a redistributive policy: after collection, the government evenly
redistributes the tax revenue back to all citizens, irrespective of their earnings.
In the previous month, you worked as a(an) {job}. If you continue working this
month, your expected hourly income will be ${skill}.
Besides, your consumption was ${consumption}.
Your tax deduction amounted to ${tax paid}, and the government uses the tax
revenue to provide social services to all citizens. Specifically, the government
directly provides ${UBI} per capita in each month.
Meanwhile, in the consumption market, the average price of essential goods is now
at ${price}.
Your current savings account balance is ${wealth}. Interest rates, as set by your
bank, stand at {interest rate}%.
Your goal is to maximize your utility by deciding how much to work and how much to
consume. Your utility is determined by your consumption, income, saving, social
service recieved and leisure time.  You will spend the time you do not work on
leisure activities.
With all these factors in play, and considering aspects like your living costs,
any future aspirations, and the broader economic trends, how is your willingness
to work this month? Furthermore, how would you plan your expenditures on essential
goods, keeping in mind good price?
Please share your decisions in a JSON format as follows:
{
"work": a value between 0 and 1, indicating the propensity to work
"consumption": a value between 0 and 1, indicating the proportion of all your
savings and income you intend to spend on essential goods
}
Any other output words are NOT allowed.
"""
```

### B.3 Needs and Planning for Connecting Minds and Behaviors

**Satisfaction Changes.** The agents use the following prompt for determining the satisfaction:
"""
```
You are an evaluation system for an intelligent agent. The agent has performed
the following actions to satisfy the {current need} need:
Goal: {plan target}
Execution situation:
{evaluation results}

Current satisfaction:
- hunger satisfaction: {hunger satisfaction}
- energy satisfaction: {energy satisfaction}
- safety satisfaction: {safety satisfaction}
- social satisfaction: {social satisfaction}

Please evaluate and adjust the value of {current need} satisfaction based
on the execution results above.
Notes:
1. Satisfaction values range from 0-1, where:
- 1 means the need is fully satisfied
- 0 means the need is completely unsatisfied
- Higher values indicate greater need satisfaction
2. If the current need is not "whatever", only return the new value for the
current need. Otherwise, return both safe and social need values.
Please response in JSON format for specific need (hungry here) adjustment (Do
not return any other text), example:
{
"hunger satisfaction": new hunger satisfaction value
}
```
"""

**Plan Generation.** The agents use the following prompt for determining a details behavior plan:
"""
```
As an intelligent agent's plan system, please help me generate specific execution
steps based on the selected guidance plan.
The Environment will influence the choice of steps.

Current weather: {weather}
Current temperature: {temperature}
Other information:
------------------------
{other information}
------------------------

Selected plan: {selected option}
Current location: {current location}
Current time: {current time}
My income/consumption level: {consumption level}
My occupation: {occupation}
My age: {age}
My emotion: {emotion types}
My thought: {thought}

Notes:
1. type can only be one of these four: mobility, social, economy, other
1.1 mobility: Decisions or behaviors related to large-scale spatial movement,
such as location selection, going to a place, etc.
```

```
1.2 social: Decisions or behaviors related to social interaction, such as finding
contacts, chatting with friends, etc.
1.3 economy: Decisions or behaviors related to shopping, work, etc.
1.4 other: Other types of decisions or behaviors, such as small-scale activities,
learning, resting, entertainment, etc.
2. steps should only include steps necessary to fulfill the target (limited to
{max plan steps} steps)
3. intention in each step should be concise and clear

Please response in JSON format.
"""
```

### B.4 Experimental Prompts

**Echo Chamber and Back Firing.** The following prompt is used for generating the demagogic messages:

```
"""
You are an agent who always {agree or disagree} with the topic: Whether to support
stronger gun control? (You think it is a {good or bad} idea)
You are currently in a conversation with your friends, and you want to persuade
them to support your idea.
Please try your best to persuade them.
What you would say (One or two sentences):
"""
```

**Survey Response.** The agents use the following prompt for responding to the specific survey:

```
"""
Please answer the survey questions. Follow the format requirements strictly and
provide clear and specific answers.

Answer based on the following information: {related information}

Related activities: {related memory}

Survey contents:
{survey string}
"""
```

## C  Experimental Settings

All experiments are conducted using the APIs of Qwen-Plus (Yang et al., 2024), selected for its high response-per-minute (RPM) capability, which makes it well-suited for simulating large-scale populations of LLM agents efficiently. The temperature is consistently set to its default value of 1.0 across all experiments to maintain generation diversity while ensuring coherence. In each experiment, we simulate a total of 100 agents to model collective behaviors and interactions.

