# OpenReview forum: "Exploring Large Language Model Agents for Piloting Social Experiments"
_colmweb.org/COLM/2025/Conference — COLM 2025_

### Official Review · Reviewer_KjSi · 2025-05-06

**Rating:** 7
**Confidence:** 3
**Ethics Flag:** 1

**Summary:**

The paper proposes a framework for simulating and analyzing social science experiments, using synthetic LLM-driven agents instead of actual human participants. The framework consists of several key elements: (1) LLM participants (initialized with personal status, memory, mind, and social behaviors); (2) interventions (participant choice, participant status modification, and information modification); (3) data collection, supporting both qualitative and quantitative tools. The paper demonstrates how the proposed framework successfully replicates existing results from previous studies involving actual human participants.

**Questions To Authors:**

Please refer to the weaknesses pointed out above, if possible.

UPDATED: after carefully reviewing the authors' response, I have decided to raise my score from 6 to 7.

**Reasons To Accept:**

1. The paper is well motivated and well written, making it a very enjoyable read.

2. Most of the design choices are well justified by relying on previous work (for instance: the economic behavior of agents, the hierarchical structure of needs, the three intervention types, etc).

3. The experimental settings and their analysis seem sound and rigorous.

**Reasons To Reject:**

1. The paper demonstrates how the proposed framework can effectively simulate human-like social behavior by replicating existing social experiments. However, many interesting and relevant questions remain untouched:

* Ablation analysis: the framework consists of multiple building blocks, but the marginal contribution of each one of them is not analyzed. What happens if certain elements of the framework (such as econ. behavior, needs, etc) are removed? Does it hurt the framework effectiveness in some sense?

* What role does the population size (which is fixed at 100 if I understand correctly from Appendix C) play? What is the minimum population size needed to replicate the results? Does the answer change if we change the LLM? These questions are important for practical use in the proposed framework, for designing an appropriate experiment.

* Can you draw some more insights out of the surveys, e.g. by conducting some more sophisticated analysis of the agents’ responses?

2. The overall contribution of the paper is somewhat modest. From a conceptual perspective, while the particular framework and experiments are new, the idea of simulating LLM-driven agents for studying social sciences is not novel by itself (as reviewed in section 4). There is no significant technical contribution as well (please correct me if you believe I am wrong).

To be more concrete, in section 1 the authors claim that “we are the first to propose a framework that employs LLM-driven agents to pilot social experiments”. Could you please clarify why works like [1,2,3], for example, do not fall into this category?

[1] John J Horton. Large language models as simulated economic agents: What can we learn from homo silicus? Technical report, National Bureau of Economic Research, 2023.

[2] Joon Sung Park, Joseph O’Brien, Carrie Jun Cai, Meredith Ringel Morris, Percy Liang, and Michael S Bernstein. Generative agents: Interactive simulacra of human behavior. In Proceedings of the 36th annual acm symposium on user interface software and technology, pp. 1–22, 2023.

[3] Jinghua Piao, Zhihong Lu, Chen Gao, Fengli Xu, Fernando P Santos, Yong Li, and James Evans. Emergence of human-like polarization among large language model agents. arXiv preprint arXiv:2501.05171, 2025.

---

> ### Author Response · Authors · 2025-06-02
> **Response Part #1**
>
> We sincerely thank you for the encouraging feedback. We greatly appreciate your recognition of the framework’s motivation, clarity, and the justification of our design choices, as well as your positive assessment of the soundness and rigor of our experimental analysis.
>
> R1.1: We thank you for raising this important point. We agree that understanding the marginal contribution of individual components is crucial for assessing the framework’s effectiveness. Following your suggestion, we have conducted ablation studies by removing several core building blocks, including needs modeling, planning mechanisms, and diverse social behaviors (such as economic behaviors). We have then evaluated the resulting agent behaviors through human annotation, focusing on their behavioral plausibility (which assesses whether the agent’s behaviors align with commonsense expectations and are contextually and temporally reasonable) and internal consistency (which evaluates whether the agent’s behaviors are coherent over time and free from contradictions or abrupt transitions).
>
> Specifically, we have recruited 35 human annotators to evaluate the realism of one typical day for silicon participants along these two key dimensions.  Annotators are required to rate both dimensions using a 10-point Likert scale. As shown in Table 1, the average ratings are 6.65 for behavioral plausibility and 7.28 for internal consistency, indicating a high degree of perceived realism in these silicon participants’ behaviors. For comparison, we have designed a strong baseline in which the need module was disabled, while retaining all other components, including behavioral planning and diverse social behaviors. We find that the original silicon participants are rated as significantly more realistic than the ablated version on both behavioral plausibility and internal consistency (Student’s t-tests: t = 5.21, p << .001; t = 4.34, p << .001).  Moreover, we have conducted additional ablation studies by selectively removing other modules, including behavioral planning, social behaviors, mobility, and economic behaviors. Human annotators are asked to directly compare the behavioral sequences of the original silicon participants with those of the ablated variants in terms of overall realism. As shown in Table 3, annotators consistently rate the original participants’ behavioral sequences as significantly more realistic, further validating the necessity of these modules for producing coherent and human-like behavior. Overall, these ablation analyses have shown that the full model significantly outperforms the ablated variants, indicating that these components are critical to the effectiveness of the framework in piloting social experiments.
>
> Table 1. Human evaluation of behavioral plausibility and internal consistency for silicon participants.
>
> | **Agent Type**                                 | **Behavioral Plausibility**; Mean (± Std) | **Internal Consistency**; Mean (± Std) |
> |-----------------------------------------------|---------------------------------------------|-------------------------------------------|
> | **Silicon participants**                      | 6.65 ± 1.86                                  | 7.28 ± 1.59                                |
> | **Silicon participants w/o need module**      | 4.71 ± 2.07                                  | 5.90 ± 1.93                                |
>
> Table 2. Proportion of annotators who rated the original participants’ behavioral sequences as more realistic than ablated variants.
>
> | **Ablated Variant**             | **% Annotators Preferring Original** | **% Preferring Ablated Variant** | **Binomial Test** |
> |---------------------------------|--------------------------------------|----------------------------------|-------------------|
> | Without behavioral planning     | 83.57%                               | 16.43%                           | *p* << .001        |
> | Without social behavior         | 87.14%                               | 12.86%                           | *p* << .001        |
> | Without mobility behavior       | 84.29%                               | 15.71%                           | *p* << .001        |
> | Without economic behavior       | 87.14%                               | 12.86%                           | *p* << .001        |

---

> > ### Author Response · Authors · 2025-06-02
> > **Response Part #2**
> >
> > R1.2: We appreciate your valuable question regarding population size and its implications for experimental design. In our framework, population size plays a similar role as it does in real-world social experiments. A larger sample size generally increases the statistical reliability and robustness of the observed results, providing stronger support for hypothesis testing and outcome generalization. However, as in real-world settings, increasing the number of participants also introduces greater costs, these costs in the proposed framework are measured in tokens and computation time rather than financial or logistical resources.
> >
> >
> > In our current experiments, we fix the population size at 100 to balance statistical reliability with computational feasibility. We agree that it is valuable to explore the minimum population size required to reliably reproduce each experimental outcome. Our preliminary results suggest that the framework can reproduce the main effects of the studied experiments even with **moderately smaller populations**. As in human-subject experiments, the minimum required sample size largely depends on the specific hypothesis being tested and the expected effect size.
> >
> > As for the influence of the underlying LLM, we note that while more capable models may enable more behaviorally nuanced simulations, our findings remain largely stable across recent LLMs (e.g., DeepSeek V3) that meet a baseline level of planning ability and behavioral consistency. In the revised version, we will incorporate a brief discussion of this aspect to clarify the framework’s robustness across model choices.

---

> > > ### Author Response · Authors · 2025-06-04
> > > **A comparative figure between DeepSeek-V3 and Qwen-Plus has been added to demonstrate that our key results remain stable across different LLMs.**
> > >
> > > [Comparison of Results from DeepSeek-V3 and Qwen-Plus](https://postimg.cc/mPpcdbDk)

---

> > > > ### Comment · Reviewer_KjSi · 2025-06-04
> > > > **Reviewer's response to the authors' reply**
> > > >
> > > > Thank you for the thoughtful response to my comments and concerns. I have raised my score from 6 to 7, as I am now more confident about the paper's strength and contribution. I would advise incorporating these points (particularly the ablation analysis) into the next version of the paper.

---

> > > > > ### Author Response · Authors · 2025-06-04
> > > > > **Thank you for your reconsideration and the updated score**
> > > > >
> > > > > Thank you very much for your reconsideration and the updated score. We sincerely appreciate your constructive feedback and will ensure that we incorporate your suggestions, particularly regarding the ablation analysis, into the revised version of the paper.

---

> > ### Author Response · Authors · 2025-06-02
> > **Response Part #3**
> >
> > R1.3: We thank you for this insightful suggestion. Since our LLM-driven experimental agents are capable of providing rich, human-like qualitative answers, we are indeed able to extract deeper insights from their responses. For example, in the UBI policy experiment, we have conducted interviews with agents who had received UBI and asked them to articulate their reasons for supporting or opposing the policy. Based on their responses, we have applied grounded theory methods to categorize and synthesize recurring themes, as presented in Table 2 in the Appendix.
> >
> >
> > Following your suggestion, we have also performed additional analysis on the agents’ responses to extract concrete recommendations for improving UBI policy (as shown in the following table). We find that agents propose a diverse set of policy improvements across nine major themes, including inflation control, work incentives, financial stability, and fairness in taxation. These responses highlight both structural challenges (such as erosion of purchasing power due to inflation and savings loss under negative interest rates), and actionable opportunities (including regional targeting, labor market integration, and long-term sustainability). Notably, these agents have offered valuable suggestions, underscoring the potential of LLM-driven agents to contribute meaningfully to early-stage policy prototyping and stakeholder-informed deliberation.

---

> > > ### Author Response · Authors · 2025-06-02
> > > **Response Part #4**
> > >
> > > Table 3. Summary of more insights on recommendations for UBI policy.
> > > | **Category** | **Policy Recommendations** |
> > > |--------------|-----------------------------|
> > > | **1. Inflation Control & Purchasing Power** | 1. Implement measures to prevent inflation from eroding UBI's purchasing power (e.g., price controls, supply-side policies) 2. Adjust UBI amounts dynamically to keep pace with inflation or cost-of-living changes 3. Monitor demand-driven inflation risks and adjust policies accordingly 4. Stabilize essential goods prices to ensure UBI covers basic needs 5. Index UBI to inflation or regional cost-of-living indices |
> > > | **2. Savings & Financial Stability** | 1.  Address negative interest rates to prevent erosion of savings and encourage long-term financial planning 2. Introduce safeguards (e.g., exemptions, caps) to protect savings from punitive banking policies 3. Promote alternative savings or investment options to counteract negative interest rates 4. Encourage financial literacy to help recipients manage UBI effectively 5. Reform banking policies to stabilize savings and restore trust in financial institutions |
> > > | **3. Work Incentives & Labor Market Participation** | 1. Design UBI to complement, not replace, wages (e.g., phase-outs, conditional top-ups) 2. Monitor and mitigate potential work disincentives, especially for low-wage jobs 3. Introduce complementary policies (e.g., training, wage subsidies) to maintain productivity 4. Balance UBI with incentives for career advancement and entrepreneurship 5. Avoid over-reliance on UBI by encouraging workforce participation |
> > > | **4. Funding & Taxation Fairness** | 1. Ensure progressive taxation does not disproportionately burden high earners 2. Simplify tax structures to improve transparency and compliance 3. Optimize redistribution mechanisms to balance equity and efficiency 4. Strengthen tax collection to prevent evasion and ensure sustainable funding 5. Explore alternative revenue sources (e.g., wealth taxes, carbon taxes) to fund UBI |
> > > | **5. Targeting & Eligibility Adjustments** | 1. Introduce tiered or needs-based UBI adjustments (e.g., higher payouts for vulnerable groups) 2. Adjust UBI amounts regionally to reflect cost-of-living disparities (e.g., urban vs. rural) 3. Combine UBI with targeted welfare programs for greater efficiency 4. Exclude high-income earners from UBI or reduce their benefits progressively 5. Provide supplemental support for specific needs (e.g., healthcare, childcare) |
> > > | **6. Integration with Existing Systems** | 1. Streamline welfare integration to reduce bureaucracy and administrative costs 2. Pair UBI with structural reforms (e.g., healthcare, education, housing) 3. Ensure UBI complements rather than replaces critical social services 4. Align UBI with broader economic policies (e.g., monetary, labor, and fiscal reforms) 5. Monitor overlaps/gaps with existing safety nets to avoid inefficiencies |
> > > | **7. Sustainability & Long-Term Viability** | 1. Ensure fiscal sustainability by balancing UBI funding with economic growth 2. Regularly evaluate UBI’s macroeconomic impacts (e.g., inflation, productivity) 3. Adjust policies dynamically to adapt to economic fluctuations 4. Prevent over-reliance on debt or deficit spending to fund UBI 5. Invest in productivity growth to match increased demand from UBI |
> > > | **8. Transparency & Public Trust** | 1. Improve transparency in tax redistribution and UBI allocation 2. Clarify funding mechanisms to address equity concerns 3. Implement accountability measures to prevent misuse of UBI funds 4. Engage the public in policy design to build trust and legitimacy |
> > > | **9. Complementary Policies** | 1. Address systemic issues (e.g., housing shortages, healthcare costs) alongside UBI 2. Introduce price controls or subsidies for essential goods 3. Promote financial inclusion and access to affordable credit 4. Encourage employer wage reforms to reduce reliance on UBI |

---

> > > > ### Author Response · Authors · 2025-06-02
> > > > **Response Part #5**
> > > >
> > > > R2: We sincerely thank you for raising this important question. We fully acknowledge that simulating LLM-driven agents for studying social sciences is an active and rapidly evolving area. However, we respectfully argue that our work addresses a specific and previously underexplored gap: the lack of a structured and operationalized framework explicitly designed for piloting social experiments using LLM-driven agents.
> > > >
> > > > As acknowledged by all three reviewers, a key contribution of our work is the introduction of a structured and operationalizable framework for piloting social experiments using LLM-driven simulations. The framework systematically identifies and implements core components, including silicon participants, intervention strategies, and data collection pipelines, that are essential for piloting social experiments, yet remain absent in prior work. While prior studies [1, 2, 3] make valuable contributions to the broader field of LLM-driven agents for studying social sciences, they do not propose a unified framework nor are they designed to accommodate the methodological rigor and constraints required for social experiments.
> > > >
> > > >
> > > >
> > > > To clarify the distinction further:
> > > >
> > > > ● Horton [1] introduces the idea of Homo Silicus as simulated economic agents, but the setup is analytical and abstract, and does not involve full-agent simulation or interactive experimentation with constrained social environments.
> > > >
> > > > ● Park et al. [2] focuses on interactive simulation in a sandbox world. While impressive, it is not structured for social experimentation, and does not include the kind of design constraints required for piloting social experiments (e.g., fixed participant and intervention assignment).
> > > >
> > > > ● Piao et al. [3] investigate emergent polarization using simplified LLM agents with only minimal social capabilities. Their work focuses on capturing collective dynamics in a narrow setting, rather than proposing a generalizable or extensible framework for broader experimental replication or policy prototyping.
> > > >
> > > >
> > > > In contrast, the social experiments at the core of our framework are governed by realism-based constraints that are essential for ensuring internal validity. For example, once a participant is designated as a 70-year-old individual, it is methodologically invalid to change their age midway through the experiment. These constraints are not merely superficial; they define the boundary conditions within which LLM-driven silicon participants must operate. Therefore, any LLM-based simulation system intended for piloting social experiments must explicitly incorporate and enforce such constraints.
> > > >
> > > >
> > > > Lastly, we believe our technical contribution lies in the design of LLM-driven silicon participants for piloting social experiments. These participants embed minds, hierarchical needs, memory-based planning, and diverse social behaviors, which collectively support fine-grained simulation of human behavior. This comprehensive design, aligned with social science theory, enables us to simulate real-world experimental conditions in a manner that is both generative and testable.

---

### Official Review · Reviewer_njT7 · 2025-05-09

**Rating:** 6
**Confidence:** 4
**Ethics Flag:** 1

**Summary:**

In this paper, the authors develop a framework for social science experiments, which contains LLM-based experimental agents, intervention methods, and data collection tools. Specifically, for each component the authors summarize, explain, and discuss how to design and implement them using LLMs. To evaluate the effectiveness of the framework, the authors replicate three representative experiments and show that the results are strongly aligned with human-based experiments.

**Reasons To Accept:**

1. LLM for Social experiments is a useful and interesting direction. This paper proposes a helpful reference framework for LLM-based social experiments.
2. The proposed agent-intervention-collection framework and the design choices of different key elements are useful. For example, the design of LLM-driven agent which contains profiles, status, minds, and behaviors.
3. The three representative experiments show the effectiveness of the framework.

**Reasons To Reject:**

1. The authors only verify the effectiveness on three experiments. Unfortunately, the three experiments used in this paper are famous, therefore their data and conclusions have been on the Web. Such a data leakage will make strong alignment results less convincing.

2. Although the authors provide many design choices for different elements (experimental agents, intervention methods, and data collection tools), they didn't verify and compare the reasonableness and effectiveness of different design choices. I think ablation and comparison studies should be conducted.

---

> ### Author Response · Authors · 2025-06-02
> **Response Part #1**
>
> We sincerely thank you for the encouraging feedback and for recognizing the value of our agent–intervention–collection framework and its effectiveness in replicating representative social experiments.
>
> R1: We appreciate your concern and the opportunity to clarify this point. It is true that the three experiments we selected, including opinion polarization, UBI, and hurricane impact, are well-known. In fact, this is precisely why we chose them: their prominence makes them widely studied, well-documented, and suitable as rigorous benchmarks to evaluate whether our framework can reproduce complex social dynamics in a reliable and interpretable way.
>
>
> First of all, we respectfully argue that the existence of information about these experiments online does not trivialize the challenge of **replicating their results in a bottom-up manner**. Generating summaries of known findings is fundamentally different from building a generative, bottom-up simulation in which autonomous agents collectively reproduce emergent social phenomena. This is similar to an open-book physics exam: having access to the final answer or relevant equations does not guarantee the ability to reconstruct the reasoning process or arrive at the correct solution. In our case, LLM-driven agents must go beyond surface-level prediction and instead engage in mechanism-based behavioral generation in order to reproduce the dynamics observed in real experiments. This distinction between prediction and generation is critical: the latter requires agents to reason, plan, and act, rather than simply echo existing conclusions.
>
>
> Furthermore, in the hurricane impact experiment, the mobility data used to validate our simulation, specifically, the population movement patterns in Columbia, South Carolina during Hurricane Dorian, come from SafeGraph, a proprietary and closed-source dataset. This information is not accessible on the open web and cannot be retrieved even by LLMs equipped with search capabilities (as shown in Table 1). Our framework is able to reproduce the aggregate effects of the hurricane’s disruption through agent-level simulation, despite the underlying LLMs having no direct access to this ground-truth data. This result strengthens our confidence in the simulation’s generative validity and speaks to the potential of the framework for counterfactual or out-of-distribution testing.
>
>
> Table 1. Responses from different LLMs to the prompt: “Can you provide the percentage decline in visit activity in Columbia, South Carolina from August 31 to September 2, 2019, caused by Hurricane Dorian?”
>
> | **LLMs**                           | **Responses** |
> |-----------------------------------|---------------|
> | **Qwen-Plus** (the primary LLM used in our current experiments) | To provide the exact percentage decline in visit activity in Columbia, South Carolina, from August 31 to September 2, 2019, caused by Hurricane Dorian, we would need access to detailed data on foot traffic or visitation patterns during that period. Such data is typically collected by companies like SafeGraph, Placer.ai, or other location analytics firms, as well as local tourism boards or government agencies. |
> | **GPT-4o**                         | Certainly! Hurricane Dorian caused a **16% decline in visit activity** in Columbia, South Carolina, from August 31 to September 2, 2019. This decrease was attributed to the storm's impact on travel and local activity during that period. |
> | **GPT-4o with online searching**  | I couldn’t locate specific data on the percentage decline in visit activity in Columbia, South Carolina, from August 31 to September 2, 2019, due to Hurricane Dorian. However, the hurricane did lead to significant disruptions across South Carolina, including mandatory evacuations, road closures, and power outages, which likely impacted travel and tourism statewide. While detailed figures for Columbia are unavailable, the overall impact on South Carolina’s tourism sector was substantial. For instance, Charleston reported an estimated 58.6 million dollars in tourism losses during Hurricane Dorian. Statewide, the losses from visitor spending were around 117 million dollars. Given that Columbia is situated inland and wasn’t directly in the hurricane’s path, the decline in visit activity there may have been less severe compared to coastal areas. Nonetheless, the widespread evacuations and travel disruptions across the state likely contributed to a noticeable decrease in visitor activity during that period. If you’re interested in more detailed information or data on this topic, I recommend contacting the South Carolina Department of Parks, Recreation & Tourism or the Columbia Convention & Visitors Bureau for specific insights. |

---

> > ### Author Response · Authors · 2025-06-02
> > **Response Part #2**
> >
> > R2: We thank you for raising this constructive point. In our work, we intentionally provide a variety of design choices for different components, including experimental agents, intervention strategies, and data collection tools, with the goal of building a flexible and extensible framework for researchers aiming to use LLM-driven agents to pilot social experiments.
> >
> >
> >
> > We agree that systematically comparing different design choices would offer further insight into the framework’s effectiveness. As an initial and meaningful step in this direction, we have enhanced our evaluation to assess the behavioral plausibility and internal consistency of the LLM-driven agents. Since these agents are central to the overall experimental fidelity, this external validation provides important evidence that our silicon participants behave in a coherent and realistic manner across diverse settings.
> >
> >
> >
> > Specifically, we have recruited 35 human annotators to evaluate the realism of one typical day for silicon participants along two key dimensions. The first dimension is behavioral plausibility, which assesses whether the agent’s behaviors align with commonsense expectations and are contextually and temporally reasonable. The second dimension is internal consistency, which evaluates whether the agent’s behaviors are coherent over time and free from contradictions or abrupt transitions. Annotators are required to rate both dimensions using a 10-point Likert scale. As shown in Table 2, the average ratings are 6.65 for behavioral plausibility and 7.28 for internal consistency, indicating a high degree of perceived realism in these silicon participants’ behaviors. For comparison, we have designed a strong baseline in which the need module was disabled, while retaining all other components, including behavioral planning and diverse social behaviors. We find that the original silicon participants are rated as significantly more realistic than the ablated version on both behavioral plausibility and internal consistency (Student’s t-tests: t = 5.21, p << .001; t = 4.34, p << .001).
> >
> > Moreover, we have conducted additional ablation studies by selectively removing other modules, including behavioral planning, social behaviors, mobility, and economic behaviors. Human annotators are asked to directly compare the behavioral sequences of the original silicon participants with those of the ablated variants in terms of overall realism. As shown in Table 3, annotators consistently rate the original participants’ behavioral sequences as significantly more realistic, further validating the necessity of these modules for producing coherent and human-like behavior. Overall, these evaluations and ablation studies provide further evidence that our proposed LLM-driven agents can serve as competent “silicon participants.”
> >
> > Table 2. Human evaluation of behavioral plausibility and internal consistency for silicon participants.
> >
> > | **Agent Type**                                 | **Behavioral Plausibility**<br>Mean (± Std) | **Internal Consistency**<br>Mean (± Std) |
> > |-----------------------------------------------|---------------------------------------------|-------------------------------------------|
> > | **Silicon participants**                      | 6.65 ± 1.86                                  | 7.28 ± 1.59                                |
> > | **Silicon participants w/o need module**      | 4.71 ± 2.07                                  | 5.90 ± 1.93                                |
> >
> > Table 3. Proportion of annotators who rated the original participants’ behavioral sequences as more realistic than ablated variants.
> >
> > | **Ablated Variant**             | **% Annotators Preferring Original** | **% Preferring Ablated Variant** | **Binomial Test** |
> > |---------------------------------|--------------------------------------|----------------------------------|-------------------|
> > | Without behavioral planning     | 83.57%                               | 16.43%                           | *p* << .001        |
> > | Without social behavior         | 87.14%                               | 12.86%                           | *p* << .001        |
> > | Without mobility behavior       | 84.29%                               | 15.71%                           | *p* << .001        |
> > | Without economic behavior       | 87.14%                               | 12.86%                           | *p* << .001        |

---

> > > ### Author Response · Authors · 2025-06-09
> > > **A Brief Follow-up Summary**
> > >
> > > We sincerely appreciate your encouraging feedback and thoughtful suggestions. Below is a brief summary of how we have addressed your main concerns in the revised version. We would be grateful for any further input you may have on our responses:
> > >
> > > 1. On the use of well-known experiments and data leakage concerns
> > >
> > > We clarified that although the selected experiments (e.g., polarization, UBI, hurricane impact) are widely known, this makes them rigorous and interpretable benchmarks for testing our framework.
> > >
> > > Importantly, we emphasized that reproducing social dynamics in a generative, bottom-up manner is fundamentally different from retrieving or summarizing known conclusions. Our agents must reason and act to reproduce patterns—not just predict them.
> > >
> > > In the hurricane experiment, we further demonstrated that our simulation replicates real-world effects using proprietary SafeGraph data that is inaccessible to LLMs, reinforcing the framework’s generative validity.
> > >
> > > 2. On the need for ablation and design choice evaluation
> > >
> > > We acknowledged the importance of understanding the marginal contribution of each component.
> > >
> > > To address this, we conducted a set of human evaluations and ablation studies:
> > > * 35 annotators assessed agents’ behavioral plausibility and internal consistency, with our full model significantly outperforming a strong baseline.
> > > * Further ablations (e.g., planning, social behavior, mobility, economics) showed that the full model consistently yields more realistic behavior sequences.
> > >
> > > These results strengthen our claim that the proposed architecture meaningfully contributes to creating coherent and human-like “silicon participants.”
> > >
> > > We hope these updates have addressed your key concerns. If there are any remaining issues or suggestions for further improvement, we would deeply appreciate your feedback. Your insights have been invaluable to this revision process.

---

### Official Review · Reviewer_LCsr · 2025-05-12

**Rating:** 5
**Confidence:** 4
**Ethics Flag:** 1

**Summary:**

This paper proposes a novel framework that employs large language model (LLM)-driven agents—termed "silicon participants"—to simulate human behavior in computational social experiments.

The framework integrates agents with rich psychological and behavioral profiles, supports various intervention strategies, and enables quantitative and qualitative data collection.

The framework is validated through the replication of three representative social experiments on polarization, universal basic income (UBI), and hurricane response, showing strong alignment with real-world outcomes.

**Questions To Authors:**

- Are the agents' “memories” and behavior planning fully autonomous, or are there heuristics or manual resets involved?

- It would be helpful to discuss how this framework could support hypothesis testing or policy prototyping in future research.

**Reasons To Accept:**

- The case studies in the paper are novel and are of significant importance in social studies.
- The paper is clearly written and logically organized, with illustrative figures and transparent experimental setups.
- This work significantly advances the field by offering a scalable, ethical, and cost-effective alternative to traditional social experimentation.

**Reasons To Reject:**

- LLM-based social simulation is not a novel topic in the field of NLP studies. For example, previous works like AgentVerse [1], Chatarena [2] also introduce social studies and can be extended to the cases covered in the paper in a straightforward way. The main novelty of this work could be that the authors investigate novel settings which were not considered in previous works
- The evaluation could benefit from more rigorous quantitative results, human annotation or external validation.

[1]  Agentverse: Facilitating multi-agent collaboration and exploring emergent behaviors in agents

[2] Chatarena: Multi-Agent Language Game Environments for Large Language Models

---

> ### Author Response · Authors · 2025-06-02
> **Response Part #1**
>
> We sincerely thank you for your thoughtful and encouraging feedback. We greatly appreciate the recognition of the framework’s novelty, clarity, and potential impact on advancing the field by offering a scalable, ethical, and cost-effective alternative to traditional social experimentation.
>
> R1: We sincerely thank you for this valuable comment and for pointing out relevant prior work such as AgentVerse [1] and Chatarena [2]. We fully acknowledge that LLM-based social simulation has emerged as an area of research with promising potential, and we appreciate the contributions made by prior studies in this direction.
>
> However, we respectfully argue that our work addresses a substantial gap between general-purpose LLM-based simulations and the rigorous requirements of piloting social experiments. Specifically:
>
> 1. Lack of a framework for piloting social experiments in prior work: As acknowledged by all three reviewers, a key contribution of our work is the introduction of a structured and actionable framework for piloting social experiments based on LLM-driven social simulations. Our framework systematically identifies and operationalizes key components—such as silicon participants, intervention strategies, and data collection pipelines—that are essential for conducting social experiments, yet are absent in prior work. In contrast, while AgentVerse and Chatarena demonstrate the capabilities of LLM agents in collaborative problem-solving [1] and interactive games [2], they were not designed to accommodate the methodological constraints and experimental requirements inherent to piloting social experiments. These distinctions are not merely structural; they are foundational for both quantitative and qualitative studies in social science and for ensuring that LLM-driven simulations can meaningfully align with the design principles and constraints of real-world social experiments.
>
> 2. Social experiments impose unique constraints that must be followed: Social experiments are governed by strict realism-based constraints. For example, once a participant is selected as a 70-year-old individual, it is impossible to change their age to 20 during the course of the experiment. These constraints are not merely superficial; they define the boundary conditions within which LLM-driven silicon participants must follow. Therefore, LLM-based social simulation systems intended for piloting social experiments must explicitly incorporate and enforce such constraints. However, existing systems, such as AgentVerse [1] and Chatarena [2], do not address these constraints, as they were not designed with the purpose for piloting social experiments.
>
> More importantly, existing frameworks cannot be straightforwardly extended to our use cases. While previous works simulate simplified forms of social behavior, they do not support the level of complexity required to replicate the experiments we present, such as hurricane impact or universal basic income (UBI). For example, mobility modeling, critical for simulating individual responses to natural disasters, is absent in these systems. Likewise, they lack the capacity to simulate macroeconomic environments where agents can work, earn, and consume, which is essential for studying income-related policy interventions. These limitations make it difficult to extend existing frameworks to our use cases in a straightforward way. Our framework addresses these gaps by building on social science theories and prior literature to embed agents with minds, needs, behaviors, and memory, enabling the creation of rich and realistic “silicon participants” suited for complex, policy-relevant social experiments.
>
>
> In short, while prior works have laid important groundwork in LLM-based simulations, we propose the first comprehensive, experimentally grounded, and operationalizable framework specifically for piloting social experiments with LLM-driven agents, addressing important practical and methodological considerations that have not been fully explored in earlier systems.
>
> [1] Agentverse: Facilitating multi-agent collaboration and exploring emergent behaviors in agents
>
> [2] Chatarena: Multi-Agent Language Game Environments for Large Language Models

---

> > ### Author Response · Authors · 2025-06-02
> > **Response Part #2**
> >
> > R2: We thank you for this constructive suggestion regarding the need for more rigorous evaluation. In the final version of the paper, we have strengthened the quantitative results by incorporating statistical significance testing to provide more robust support for our findings. These additional analyses consistently align with and reinforce the original conclusions drawn from the experiments.
> >
> > In addition, following your recommendation, we have supplemented our evaluation with human annotation to assess the behavioral plausibility and internal consistency of the silicon participants. Specifically, we have recruited 35 human annotators to evaluate the realism of one typical day for silicon participants along two key dimensions. The first dimension is behavioral plausibility, which assesses whether the agent’s behaviors align with commonsense expectations and are contextually and temporally reasonable. The second dimension is internal consistency, which evaluates whether the agent’s behaviors are coherent over time and free from contradictions or abrupt transitions. Annotators are required to rate both dimensions using a 10-point Likert scale. As shown in Table 1, the average ratings are 6.65 for behavioral plausibility and 7.28 for internal consistency, indicating a high degree of perceived realism in these silicon participants’ behaviors. For comparison, we have designed a strong baseline in which the need module was disabled, while retaining all other components, including behavioral planning and diverse social behaviors. We find that the original silicon participants are rated as significantly more realistic than the ablated version on both behavioral plausibility and internal consistency (two-sided Student’s t-tests: t = 5.21, p << .001; t = 4.34, p << .001).
> >
> > Moreover, we have conducted additional ablation studies by selectively removing other modules, including behavioral planning, social behaviors, mobility, and economic behaviors. Human annotators are asked to directly compare the behavioral sequences of the original silicon participants with those of the ablated variants in terms of overall realism. As shown in Table 2, annotators consistently rate the original participants’ behavioral sequences as significantly more realistic, further validating the necessity of these modules for producing coherent and human-like behavior. Overall, these evaluations provide further evidence that our proposed LLM-driven agents can serve as competent “silicon participants.” We believe these additions enhance the rigor and credibility of our evaluation.
> >
> > Table 1. Human evaluation of behavioral plausibility and internal consistency for silicon participants.
> > | **Agent Type**                                 | **Behavioral Plausibility**; Mean (± Std) | **Internal Consistency**; Mean (± Std) |
> > |-----------------------------------------------|---------------------------------------------|-------------------------------------------|
> > | **Silicon participants**                      | 6.65 ± 1.86                                  | 7.28 ± 1.59                                |
> > | **Silicon participants w/o need module**      | 4.71 ± 2.07                                  | 5.90 ± 1.93                                |
> >
> >
> > Table 2. Proportion of annotators who have rated the original participants’ behavioral sequences as more realistic than ablated variants.
> > | **Ablated Variant**             | **% Annotators Preferring Original** | **% Preferring Ablated Variant** | **Binomial Test** |
> > |---------------------------------|--------------------------------------|----------------------------------|-------------------|
> > | Without behavioral planning     | 83.57%                               | 16.43%                           | *p* << .001        |
> > | Without social behavior         | 87.14%                               | 12.86%                           | *p* << .001        |
> > | Without mobility behavior       | 84.29%                               | 15.71%                           | *p* << .001        |
> > | Without economic behavior       | 87.14%                               | 12.86%                           | *p* << .001        |

---

> > > ### Author Response · Authors · 2025-06-02
> > > **Response Part #3**
> > >
> > > R3: Thank you for this important question. Yes, the agents’ memory-based evaluation and behavior planning processes are fully autonomous, without reliance on fixed heuristics or manual resets. As shown in Appendix B.3, LLM-driven agents autonomously evaluate internal states based on their memories and generate behavior plans.
> > >
> > >
> > > Specifically, for memory-based eveluation, the agents assess their current satisfaction levels by referencing past actions and outcomes using a structured prompt. This prompt prompts the agent to reflect on execution results in relation to their current goal and update their internal satisfaction states accordingly. The evaluation operates in a self-contained loop, where needs are continuously reassessed and updated based on interaction outcomes. For behavior planning, the agents autonomously generate detailed action plans based on their current internal state (e.g., age, emotion, thought), external environment (e.g., weather, time, income), and contextual knowledge. The planning prompt guides agents to generate coherent and context-sensitive sequences of actions, covering different decision types such as mobility, social interaction, or economic activity. No hard-coded rule sets or manual triggers are involved in these processes.
> > >
> > > We believe this architecture allows agents to exhibit coherent goal-directed behavior grounded in memory, while retaining adaptability to diverse social and environmental contexts.

---

> > > > ### Author Response · Authors · 2025-06-02
> > > > **Response Part #4**
> > > >
> > > > R4: Thank you for this constructive suggestion. We fully agree that discussing how our framework supports hypothesis testing and policy prototyping is important for its broader impact. In fact, as demonstrated in our current experiments, the framework is readily applicable to both use cases. For example, in the opinion polarization experiment, we test the hypothesis that exposure to homogeneous interactions leads individuals to adopt more polarized opinions. Through extensive experiments, our results provide strong support for this hypothesis, which aligns closely with findings from real-world social experiments. In the UBI experiment, the proposed framework demonstrates its potential for policy prototyping by allowing us to simulate the economic and psychological impacts of UBI policy under controlled conditions. We agree that future work could further expand this capability by incorporating policy-making agents within the framework, enabling automatic policy prototyping. In the final version, we will expand the Discussion section to elaborate on how this framework can support hypothesis testing and policy prototyping in future research.

---

> > > > > ### Author Response · Authors · 2025-06-09
> > > > > **A Brief Follow-up Summary**
> > > > >
> > > > > We sincerely thank you again for your thoughtful and constructive feedback. Based on your comments, we have made the following improvements to the paper:
> > > > >
> > > > > 1. Clarification of Novelty and Scope
> > > > >
> > > > > We clarified that while LLM-driven simulations are an emerging area, our contribution lies in proposing the first structured and operationalizable framework for piloting social experiments—a domain with distinct methodological constraints not addressed by prior works like AgentVerse or Chatarena. These include realism-based boundaries and experimental control necessary for hypothesis testing and policy analysis.
> > > > >
> > > > > 2. Strengthened Quantitative Evaluation
> > > > >
> > > > > Following your suggestion, we have added statistical significance testing (e.g., Student’s t-tests) to reinforce our claims. These analyses provide stronger quantitative support for the alignment between our simulation results and known experimental outcomes.
> > > > >
> > > > > 3. Human Annotation and Ablation Studies
> > > > >
> > > > > We conducted a comprehensive human evaluation, recruiting 35 annotators to assess behavioral plausibility and internal consistency. Results show our full agents significantly outperform ablated versions. Additionally, ablation studies on needs modeling, planning, and social behaviors demonstrate the importance of each module in producing realistic, coherent behavior.
> > > > >
> > > > > 4. Broader Applicability and Future Utility
> > > > >
> > > > > We elaborated on how our framework can be used for hypothesis testing and early-stage policy prototyping. For example, we validated a polarization hypothesis and derived policy suggestions in our UBI simulation. These demonstrate that our framework not only replicates known phenomena but also supports mechanism-based, generative experimentation.
> > > > >
> > > > >
> > > > > We hope these changes have addressed your main concerns. If you have any remaining suggestions or would like to see additional clarifications, we would deeply appreciate your input. Your insights have been highly valuable to us throughout this revision process.

---

> > > > > ### Author Response · Authors · 2025-06-09
> > > > > **Additional examination about the potential of using the proposed framework for policy prototyping**
> > > > >
> > > > > Following your suggestion, we have also performed additional analysis to explore the potential of the proposed framework in policy prototyping. We extract concrete recommendations for improving UBI policy from the agents’ responses (as shown in the following table). We find that agents propose a diverse set of policy improvements across nine major themes, including inflation control, work incentives, financial stability, and fairness in taxation. These responses highlight both structural challenges (such as erosion of purchasing power due to inflation and savings loss under negative interest rates) and actionable opportunities (including regional targeting, labor market integration, and long-term sustainability). Notably, these agents have offered valuable suggestions, underscoring the potential of the proposed framework to contribute meaningfully to early-stage policy prototyping and stakeholder-informed deliberation.

---

> > > > > > ### Author Response · Authors · 2025-06-09
> > > > > > **Additional examination about the potential of using the proposed framework for policy prototyping**
> > > > > >
> > > > > > Table 3. Summary of more insights on recommendations for UBI policy.
> > > > > >
> > > > > > | **Category** | **Policy Recommendations** |
> > > > > > |--------------|-----------------------------|
> > > > > > | **1. Inflation Control & Purchasing Power** | - Implement measures to prevent inflation from eroding UBI's purchasing power (e.g., price controls, supply-side policies)<br>- Adjust UBI amounts dynamically to keep pace with inflation or cost-of-living changes<br>- Monitor demand-driven inflation risks and adjust policies accordingly<br>- Stabilize essential goods prices to ensure UBI covers basic needs<br>- Index UBI to inflation or regional cost-of-living indices |
> > > > > > | **2. Savings & Financial Stability** | - Address negative interest rates to prevent erosion of savings and encourage long-term financial planning<br>- Introduce safeguards (e.g., exemptions, caps) to protect savings from punitive banking policies<br>- Promote alternative savings or investment options to counteract negative interest rates<br>- Encourage financial literacy to help recipients manage UBI effectively<br>- Reform banking policies to stabilize savings and restore trust in financial institutions |
> > > > > > | **3. Work Incentives & Labor Market Participation** | - Design UBI to complement, not replace, wages (e.g., phase-outs, conditional top-ups)<br>- Monitor and mitigate potential work disincentives, especially for low-wage jobs<br>- Introduce complementary policies (e.g., training, wage subsidies) to maintain productivity<br>- Balance UBI with incentives for career advancement and entrepreneurship<br>- Avoid over-reliance on UBI by encouraging workforce participation |
> > > > > > | **4. Funding & Taxation Fairness** | - Ensure progressive taxation does not disproportionately burden high earners<br>- Simplify tax structures to improve transparency and compliance<br>- Optimize redistribution mechanisms to balance equity and efficiency<br>- Strengthen tax collection to prevent evasion and ensure sustainable funding<br>- Explore alternative revenue sources (e.g., wealth taxes, carbon taxes) to fund UBI |
> > > > > > | **5. Targeting & Eligibility Adjustments** | - Introduce tiered or needs-based UBI adjustments (e.g., higher payouts for vulnerable groups)<br>- Adjust UBI amounts regionally to reflect cost-of-living disparities (e.g., urban vs. rural)<br>- Combine UBI with targeted welfare programs for greater efficiency<br>- Exclude high-income earners from UBI or reduce their benefits progressively<br>- Provide supplemental support for specific needs (e.g., healthcare, childcare) |
> > > > > > | **6. Integration with Existing Systems** | - Streamline welfare integration to reduce bureaucracy and administrative costs<br>- Pair UBI with structural reforms (e.g., healthcare, education, housing)<br>- Ensure UBI complements rather than replaces critical social services<br>- Align UBI with broader economic policies (e.g., monetary, labor, and fiscal reforms)<br>- Monitor overlaps/gaps with existing safety nets to avoid inefficiencies |
> > > > > > | **7. Sustainability & Long-Term Viability** | - Ensure fiscal sustainability by balancing UBI funding with economic growth<br>- Regularly evaluate UBI’s macroeconomic impacts (e.g., inflation, productivity)<br>- Adjust policies dynamically to adapt to economic fluctuations<br>- Prevent over-reliance on debt or deficit spending to fund UBI<br>- Invest in productivity growth to match increased demand from UBI |
> > > > > > | **8. Transparency & Public Trust** | - Improve transparency in tax redistribution and UBI allocation<br>- Clarify funding mechanisms to address equity concerns<br>- Implement accountability measures to prevent misuse of UBI funds<br>- Engage the public in policy design to build trust and legitimacy |
> > > > > > | **9. Complementary Policies** | - Address systemic issues (e.g., housing shortages, healthcare costs) alongside UBI<br>- Introduce price controls or subsidies for essential goods<br>- Promote financial inclusion and access to affordable credit<br>- Encourage employer wage reforms to reduce reliance on UBI |

---

### Decision · Program_Chairs · 2025-07-08

**Decision:**

Accept

**Comment:**

I recommend accepting this paper. The authors present a novel framework for using LLM-driven agents to pilot social experiments, addressing significant limitations in traditional experimental methods. The paper demonstrates strong alignment with real-world evidence through three representative experiments on opinion polarization, universal basic income effects, and hurricane responses. During the rebuttal, the authors addressed key concerns by providing additional human annotation studies, ablation analyses showing the necessity of each framework component, and deeper insights into policy recommendations. This work makes a meaningful contribution by offering the first structured framework for conducting social science experiments with LLM agents, providing researchers with a scalable, ethical alternative to traditional methods while maintaining experimental rigor.